# Enrichment of Features for Classification Using an Optimized Linear/Non-Linear Combination of Input Features

## Abstract

Automatic classification of objects is one of the most important tasks in engineering and data mining applications. Although using more complex and advanced classifiers can help to improve the accuracy of classification systems, it can be done by analyzing data sets and their features for a particular problem. Feature combination is the one which can improve the quality of the features. In this paper, a structure similar to Feed-Forward Neural Network (FFNN) is used to generate an optimized linear or non-linear combination of features for classification. Genetic Algorithm (GA) is applied to update weights and biases. Since nature of data sets and their features impact on the effectiveness of combination and classification system, linear and non-linear activation functions (or transfer function) are used to achieve more reliable system. Experiments of several UCI data sets and using minimum distance classifier as a simple classifier indicate that proposed linear and non-linear intelligent FFNN-based feature combination can present more reliable and promising results. By using such a feature combination method, there is no need to use more powerful and complex classifier anymore.

**keywords:** Classification, Feature Combination, Feature Mapping, Feed-Forward Neural Network, Genetic Algorithm, Linear Transfer Function, Non-Linear Transfer Function.

## 1 Introduction

A quick review of engineering problems reveals importance of classification and its application in medicine, mechanical and electrical engineering, computer science, power systems and so on. Some of its important applications include disease diagnosis using classification methods to diagnosis Thyroid (Temurtas (2009)), Parkinson (Das (2010)) and Alzheimers disease (Górriz et al. (2011)); or fault detection in power systems such as (Gketsis et al. (2009)) which uses classification methods to detect winding fault in windmill generators;(Palmero et al. (2005)) using neuro-fuzzy based classification method to detect faults in AC motor; and also fault detection in batch processes in chemical engineering (Zhou et al. (2003)). In all classification problems extracting useful knowledge and features from data such as image, signal, waveform and etcetera can lead to design efficient classification systems. As extracted data and their features are not usually suitable for classification purpose, two major approaches can be substituted. First approach considers all the classifiers and tries to select effective ones, even if their complexity and computational cost are increased. Second approach focusing on the features, enhances the severability of data, and then uses improved features and data for classification.

Feature combination is one of the common actions used to enhance features. In classic combination methods, deferent features vectors are lumped into a single long composite vector (Yin et al. (2005)). In some modern techniques, in addition to combination of feature vectors, dimension of feature space is reduced. Reduction process can be done by feature selection, transmission, and projection or mapping techniques, such as Linear Discriminate Analysis (LDA), Principle Component Analysis (PCA), Independent Component Analysis (ICA) and boosting (Yin et al. (2005)). In more applications, feature combination is fulfilled to improve the efficiency of classification system such as (Das et al. (2012)), that PCA and Modular PCA (MPCA) along Quad-Tree based hierarchically derived Longest Run (QTLR) features are used to recognize handwritten numerals as a statistical-

topological features combination. The other application of feature combination is used for English character recognition, here structure and statistical features combine then BP network is used as a classifier (Yang et al. (2011)). Feature combination has many applications; however before using, some questions should be answered: which kind of combination methods is useful for studied application and available data set. Is reduction of feature space dimension always useful? Is linear feature combination method better than non-linear one?

In this paper, using structure of Feed-Forward Neural Network (FFNN) along with Genetic Algorithm (GA) as a powerful optimization algorithm, Linear Intelligent Feature Combination (LIFC) and Non-Linear Intelligent Feature Combination (NLIFC) systems is introduced to present adaptive combination systems with the nature of data sets and their features. In proposed method, original features are fed into semi-FFNN structure to map features into new feature space, and then outputs of this intelligent mapping structure are classified by minimum distance classifier via cross-validation technique. In each generation, weights and biases of semi-FFNN structure are updated by GA and correct recognition rate (or error recognition rate) is evaluated.

In the rest of this paper, overview of minimum distance classifier, Feed-Forward Neural Network structure and Genetic Algorithm are described in sections2, 3and 4, respectively. In section 5, proposed method and its mathematical consideration are presented. Experimental results, comparison between proposed method and other feature combinations and classifiers using the same database are discussed in section 6. Eventually, conclusion is presented in section 7.

## 2 OVERVIEW OF MINIMUM DISTANCE CLASSIFIER

*Minimum Distance classifier* (or 1-nearest neighbor classifier) is one of the simplest classification methods, which works based on measured distance between an unknown input data and available data in classes. Distance is defined as an index of similarity, according this definition, the minimum distance means the maximum similarity. Distance between two vectors can be calculated in various procedures, such as Euclidian distance, Normalized Euclidian distance, Mahalanobis distance, Manhattan distance and etcetera. Euclidian distance is the most prevalent procedure that is presented in 1.

$$D = | \, |X - Y| \, | = \sqrt{\sum_{i=1}^{n} (x_i - y_i)^2} \tag{1}$$

Where $D$ is the distance between two vectors $X$ and $Y$. $||X - Y||$ means second norm of Euclidian distance. Notation $n$ is dimension of $X$ and $Y$ where $X = (x_1, x_2, , x_n)$ and $Y = (y_1, y_2, , y_n)$. Fig.1 shows the concept of a minimum distance classifier. As it can be seen, distance between unknown input data and $C2$ is the minimum distance among all distances therefore this input data assigns to class $C$.

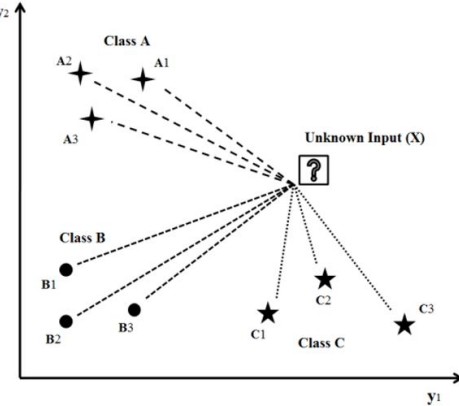

Figure 1: An example of a minimum distance classifier.

## 3 OVERVIEW OF FEED-FORWARD NEURAL NETWORK STRUCTURE

Artificial Neural Networks (ANNs) are designed based on a model of human brain and its neural cells. Although human knowledge is much more limited than brain; its performance can be understood according to observation and physiology and anatomy information of brain (Ramsden (2003)). Prominent trait of ANN is its ability to learn complicated problems between input and output vectors. In general, these networks are capable to model many non-linear functions. This ability lets neural networks be used in practical problems such as comparative diagnoses and controlling non-linear systems. Nowadays, different topologies are proposed for implementing ANN in supervised, unsupervised and reinforcement applications. Feed-forward is a dominant used topology in supervised learning procedure. Feed-forward topology for an ANN is shown in Fig. 2. As it can be seen, information is fed into ANN via input layer which distribute just input information into the main body of ANN. In this transmission the quantities of information are changed through multiplying by synapse weights of connection between input layer and next layer. Applying activation functions in next layers, updated information arrive at output layer. General equation is given by 2. It is noticeable that in this structure information flow from input to output and there is not any feedback, also there is not any disconnection and jump connection between layers.

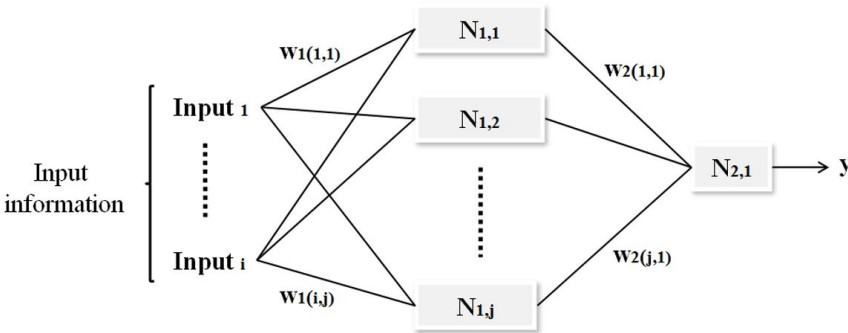

Figure 2: A typical structure of feed-forward topology for neural network .

$$y = s[\sum_{k=1}^{j} w_2(k,1) \cdot (\sum_{l=1}^{i} g[w_1(l,k) \cdot Input_i])] \tag{2}$$

Where, coefficients $g$ and $s$ are activation functions of $N2$ and $N1$s, respectively. $w1$s are synapse weights between input layer and hidden layer, also $w2$s are synapse weights between hidden layer and output layer.

## 4 OVERVIEW OF GENETIC ALGORITHM

In evolution theory, particles of population evolve themselves to be more adaptable to their environment. Therefore the particles that can do this better have more chance to survive. These algorithms are stochastic optimization techniques. In this kind of techniques, information of each generation is transferred to next generation by chromosome. Each chromosome consists of gens and any gen illustrates an especial feature or behavior.

Genetic Algorithm (GA) is one of the most well known evolutionary algorithms. In GA's process, first of all, initial population is created based on necessities of problem. After that, objective function is evaluated. In order to achieve the best solution, off springs are created from parents in reproduction step by crossover and mutation. Consequently the best solution is obtained after determined iterations (Melanie (1999)).

## 5 PROPOSED METHOD AND ITS MATHEMATICAL CONSIDERATION

As mentioned before, variant methods may be used to improve the ability of classification system. In some cases, we interest in more complex and powerful classifier, although helpful, it often reduces decision making speed and increases computational cost. The other way is using pre-processing on training data before changing the kind of classifier or its complexity. Feature combination is one of the most common ways used to enhance the quality of features, so simple classifiers can discriminate them easily. In the most feature combination methods, such as LDA, PCA, ICA, MPCA and etcetera the main strategy is to reduce the feature space dimension, whereas based on nature of data sets features sometimes dimension reduction is needed, combination of features in same dimension is enough sometimes, and also increase of feature space dimension may be useful sometimes.

The main idea of proposed method in this paper is to applying linear or non-linear intelligent features map in new solution space, in this method discriminative of data is increased. In general, proposed method is illustrated as Fig. 3 and can be represented as follow:

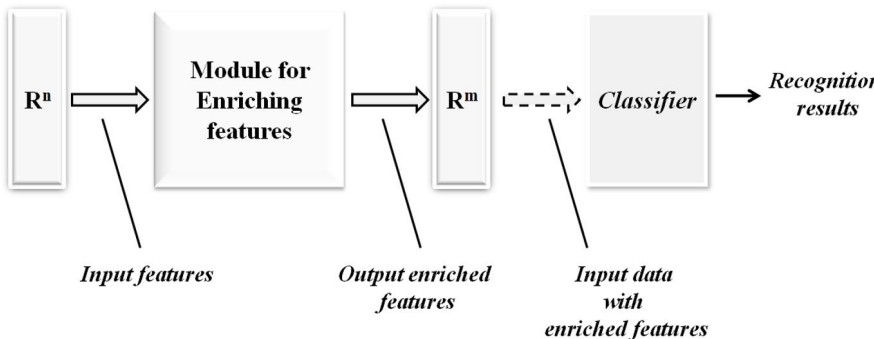

Figure 3: block diagram of proposed method to enrich futures and then use them for classification .

Let $R$ be solution space; according to the mapping concepts, we have:

$$f : R^n \rightarrow R^m \tag{3}$$

Where, Superscripts $n$ and $m$ are dimensions of solution space (or feature dimensions) before and after mapping process respectively. If $n > m$, then feature dimension is reduced from n-dimension to m-dimension by transfer function $f$. If $n = m$, then there is not any change dimensionality and only transfer function is applied on features. Feature dimension is also increased for $n < m$.

Equation 1 describes the only generality of issue, whereas in proposed method not only the feature space dimension is changed and transfer function is applied, but also features are combined in linear or non-linear format. As shown in Fig. 4 $X = x_1, x_2, , x_n$ is an input data, $Y = y_1, y_2, , y_m$ is an output data and $F$ is a transfer function which can be a typical Super polynomial like a feed-forward neural network structure.

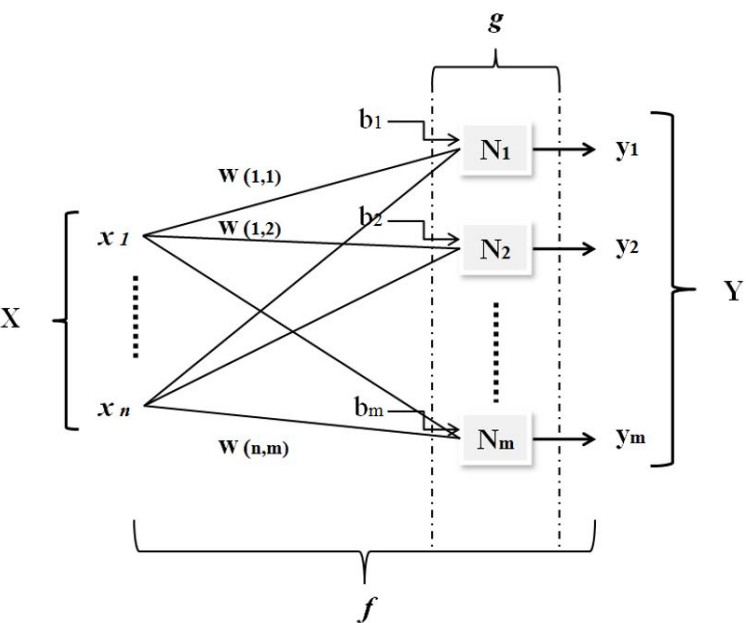

Figure 4: A typical scheme of intelligent mapping and combining system.

After feeding $X$ into this structure, if m be a unit ($m = 1$), we would project all features into one axis (or one dimension), and so we have:

$$Y = y_1 = g[(\sum_{l=1}^{n} w(l,1) \cdot x_l) + b_1] \tag{4}$$

And if $m$ be more than unit ($m > 1$), then we have:

$$Y = \begin{cases} y_1 = g[(\sum_{l=1}^{n} w(l,1) \cdot x_l) + b_1] \\ y_2 = g[(\sum_{l=1}^{n} w(l,2) \cdot x_l) + b_2] \\ \vdots \\ y_m = g[(\sum_{l=1}^{n} w(l,m) \cdot x_l) + b_m] \end{cases} \tag{5}$$

Where function g may be linear or non-linear activation (or transfer) function. In this paper, as it can be seen in Fig. 5 (a) and Eq. 6 in the case of linear transfer function (like purelin) $y_s$ are only the weighted summation of primary features ($x_s$).

$$g = x \tag{6}$$

Function $g$ can be non-linear as shown in Eq. 7and Fig. 5 (b). This non-linear function is a kind of sigmoid transfer function which can be changed by coefficient $\alpha$ .

$$g = \frac{2}{1 + e^{-\alpha x}} + 1 \tag{7}$$

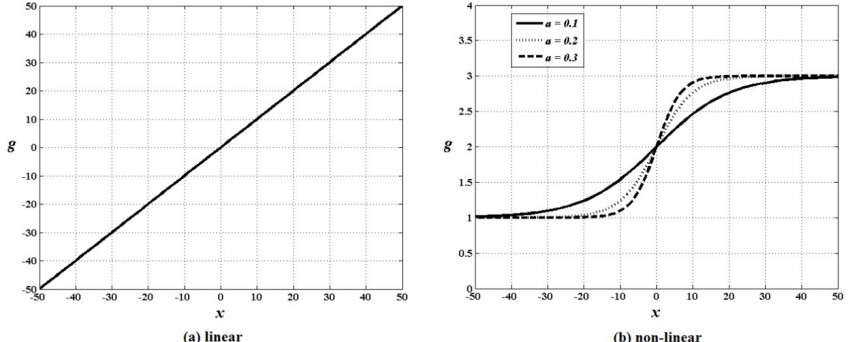

Figure 5: Linear and non-linear used activation function in proposed method.

Now finding the optimum weights and biases for increasing separability of features that leads to increase efficiency of classification system is the heart of this paper. GA is utilized as one of the most accurate and powerful optimization tool. It is applied on features of each data after establishing the structure of mentioned intelligent mapping system and considering weights and biases with initial random value as shown in Fig. 6.Then a very simple classifiers used that is minimum distance classifier and cross-validation technique - one-leave one-out (Bishop (2006)) and error recognition rate is calculated. In this step, stop criteria is evaluated which is the least error recognition rate or the given number of generation for GA. If neither of stop criterions is satisfied, GA updates weights and biases. This process is done again and again until one of the stop criterions become satisfied. In other word mapping system (intelligent combination system) is a fitness function of GA and weights, and biases are GAs chromosomes.

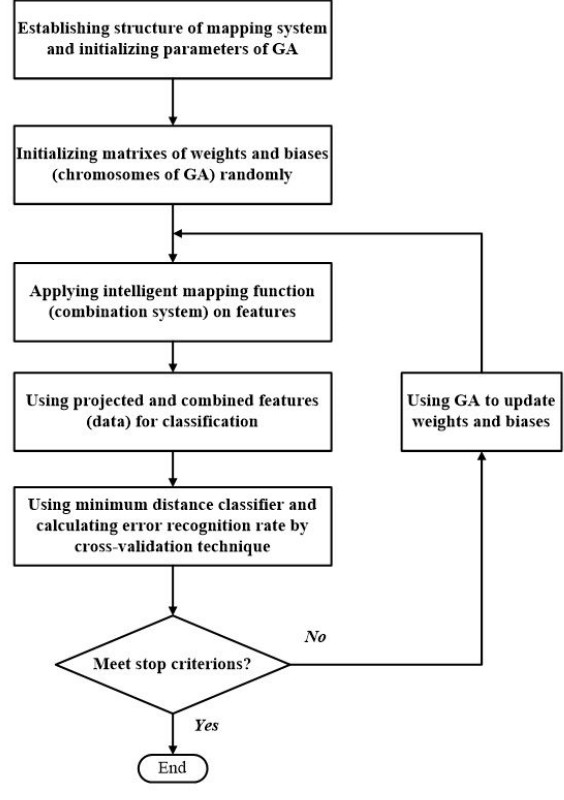

Figure 6: Flowchart of proposed intelligent combination system.

# 6 EXPERIMENT

In order to evaluate the ability of proposed combination methods four classification tasks of UCI data sets are used (Blake & Merz (1998)). Useful information of studied data sets [1] is described as follow (see 1).

*Iris:* The Iris data contains 50 samples from three species, namely, Iris setosa, Iris versicolor, and Iris virginica.Sepal length, sepal width, petal length and petal width are four features extracted from each species.

*Wine:* These data are the results of a chemical analysis of wines grown in the same region in Italy and they are derived from three different cultivars. The analysis determines the quantities of 13 features extracted from each type of wine. These features are Alcohol, Malic acid, Ash, Alcalinity of Ash, Magnesium, total phenols, flavanoids, non-flavanoid phenols, proanthocyanins, color intensity, hue OD289/OD315 of diluted wines, and proline.Moreover, this dataset contains 178 samples categorized in three classes.

*Glass:* The Glass data set consists of 214 samples of nine features from every specie: building-windows-float-processed, building-windows-non-float-processed, vehicle-windows-float-processed, containers, tableware, and heal ware.And extracted features are Refractive Index, Sodium, Magnesium, Aluminum, Silicon, Potassium, Calcium, Barium, and Iron.

*Ionosphere:* This radar data was collected by a system in Goose Bay, Labrador. This system consists of a phased array of 16 high-frequency antennas with a total transmitted power on the order of 6.4 kilowatts. This radar data are categorized in two groups; "Good" and "Bad". "Good" radar returns show evidence of some types of structure in the ionosphere. And "Bad" returns dont do so, their signals pass through the ionosphere.

Table 1: University of California Irvine (UCI) datasets used in the experiments

| Datasets | Number of Class | Dimension of Features | Number of Samples |
|---|---|---|---|
| IRIS | 3 | 4 | 150 |
| WINE | 3 | 13 | 178 |
| GLASS | 6 | 9 | 214 |
| IONOSPHERE | 2 | 34 | 351 |

As mentioned before, studying the nature of data set in order to design efficient combination and classification systems may be so important. Therefore all possible condition (namely: dimension reduction, dimension increase, only combination of features in same dimension, linear and non-linear mapping) are considered. For each data set, classification using cross-validation is applied ten times. Classification parameters of GA are also considered similar for all data sets to present same condition, as shown in Table (2).Coefficient $\alpha$ is 0.2for all non-linear feature combinations. Fig.7 shows typical convergence curves of error recognition rate for studied data sets.

Table 2: consider parameters for Genetic Algorithm which is same in all tests.

| | |
|---|---|
| Population size | 50 |
| Selection function | Roulette |
| Scaling function | Rank |
| Mutation function | Gaussian |
| Crossover function | Two point |

---

[1]These data sets are available from the site:http://archive.ics.uci.edu/ml/datasets

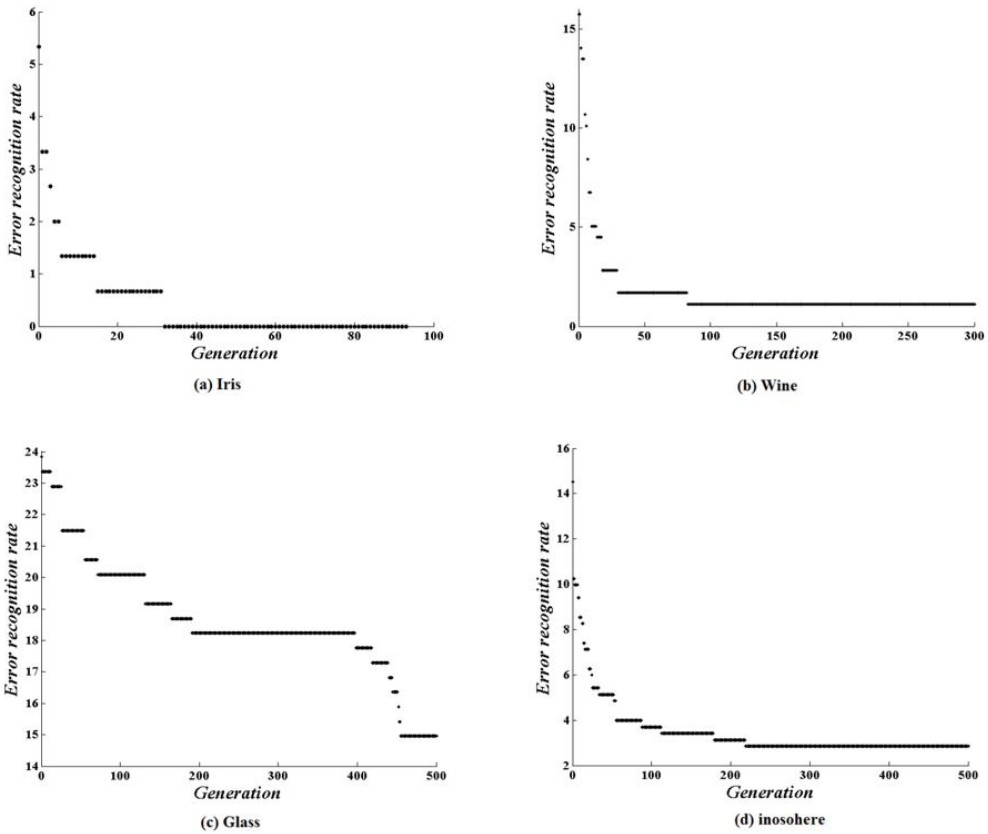

Figure 7: Typical convergence curves of error recognition rates for all studied data set of UCI using GA. It is worth noting that; *Correct recognition rate = 100  Error recognition rate*.

Tables (3) and (4) present obtained results from all studied data sets in all mentioned condition. In each condition, classification is done 10 times. Minimum, maximum and average of correct classification rates are calculated to evaluate accuracy and reliability. It should be mentioned that proposed method works based on error recognition rate, but we report correct recognition rate, here. In order to clear this Tables, consider Iris which has 4 features or dimensions. In first condition features are projected and combined into lower dimension (with 2 dimensions). In second, only combination of features is fulfilled under intelligent combination function and in third one, features are projected and combined into higher dimension (with 8 dimensions). In all conditions, classification is done using linear (L) and non-linear (NL) transfer function.

As it can be implied from Tables (3) and (4), best combination form for Iris is non-linear mapping that dimension of feature space is reduced, although obtained results in different condition are approximately similar winner condition is more reliable and accurate. The best classification rate for Wine is obtained by non-linear combination method while feature space dimension is increased. It is completely different for Glass in order to achieve efficient classification system for Glass it is enough to combine features without any changing in feature space dimension. Also, using non-linear combination feature while dimension is reduced can lead to best recognition rates for Ionosphere.

## 6.1 OBTAINED RESULTS IN COMPARISON WITH RESULT OF OTHER CLASSIFICATION METHODS

In order to compare the performance of proposed method with other combination methods, two common used combination methods, LDA and PCA, are considered in this section. Both methods reduce dimension of feature space. LDA reduce dimension of features to (C-1) dimensions which C is the number of classes. PCA is also reduced feature dimension, but in PCA projected feature

Table 3: Obtained results for Iris and Wine for 10 times classification in 6 conditions.

| | Iris ($R^4$) | | | | | | Wine ($R^{13}$) | | | | | |
| | $R^2$ | | $R^4$ | | $R^8$ | | $R^6$ | | $R^{13}$ | | $R^{26}$ | |
| TESTS | L (%) | NL (%) | L (%) | NL (%) | L (%) | NL (%) | L (%) | NL (%) | L (%) | NL (%) | L (%) | NL (%) |
|---|---|---|---|---|---|---|---|---|---|---|---|---|
| 1 | 100 | 100 | 100 | 100 | 98.67 | 99.34 | 97.20 | 97.20 | 96.63 | 94.95 | 95.50 | 97.76 |
| 2 | 99.34 | 100 | 100 | 100 | 100 | 100 | 95.50 | 96.63 | 95.50 | 98.87 | 91.58 | 98.31 |
| 3 | 99.34 | 100 | 100 | 100 | 100 | 99.34 | 97.20 | 96.63 | 93.26 | 97.20 | 93.26 | 99.43 |
| 4 | 100 | 100 | 100 | 100 | 99.34 | 100 | 96.07 | 97.75 | 93.26 | 95.50 | 96.07 | 97.75 |
| 5 | 100 | 100 | 99.34 | 100 | 99.34 | 100 | 96.63 | 98.87 | 93.82 | 98.87 | 96.63 | 96.63 |
| 6 | 100 | 100 | 100 | 100 | 98.67 | 99.34 | 97.75 | 97.75 | 95.50 | 98.87 | 95.50 | 99.43 |
| 7 | 100 | 100 | 98.67 | 100 | 99.34 | 100 | 97.20 | 98.87 | 97.75 | 99.43 | 97.75 | 98.87 |
| 8 | 100 | 100 | 99.34 | 100 | 100 | 100 | 95.50 | 98.31 | 97.20 | 98.31 | 97.20 | 97.75 |
| 9 | 100 | 100 | 100 | 100 | 99.34 | 100 | 95.50 | 98.87 | 96.63 | 96.63 | 97.75 | 98.31 |
| 10 | 99.34 | 100 | 100 | 99.34 | 99.34 | 99.34 | 97.75 | 96.63 | 95.50 | 98.87 | 96.63 | 99.43 |
| Min. | 99.34 | 100 | 98.67 | 99.34 | 98.67 | 99.34 | 95.50 | 96.63 | 93.26 | 94.95 | 91.58 | 96.63 |
| Max. | 100 | 100 | 100 | 100 | 100 | 100 | 97.75 | 98.87 | 97.75 | 99.43 | 97.75 | 99.43 |
| Avg. | 99.80 | 100 | 99.73 | 99.93 | 99.40 | 99.73 | 96.63 | 97.75 | 95.50 | 97.75 | 95.78 | 98.36 |

Table 4: Obtained results for Iris and Wine for 10 times classification in 6 conditions.

| | Glass ($R^9$) | | | | | | Ionosphere ($R^{34}$) | | | | | |
| | $R^5$ | | $R^9$ | | $R^{18}$ | | $R^{10}$ | | $R^{34}$ | | $R^{64}$ | |
| TESTS | L (%) | NL (%) | L (%) | NL (%) | L (%) | NL (%) | L (%) | NL (%) | L (%) | NL (%) | L (%) | NL (%) |
|---|---|---|---|---|---|---|---|---|---|---|---|---|
| 1 | 85.04 | 80.11 | 82.24 | 79.44 | 84.11 | 81.31 | 96.58 | 96.01 | 94.3 | 94.59 | 93.45 | 94.87 |
| 2 | 79.90 | 79.44 | 85.04 | 80.85 | 82.71 | 82.71 | 95.16 | 96.58 | 94.3 | 95.16 | 94.59 | 94.02 |
| 3 | 82.71 | 77.57 | 82.24 | 78.51 | 80.01 | 80.37 | 94.87 | 96.29 | 94.02 | 92.87 | 93.16 | 94.59 |
| 4 | 79.90 | 79.90 | 81.31 | 79.44 | 81.31 | 79.44 | 94.59 | 95.72 | 94.02 | 95.16 | 93.16 | 94.02 |
| 5 | 82.24 | 79.44 | 86.45 | 81.77 | 80.84 | 82.24 | 96.58 | 97.15 | 92.88 | 93.45 | 95.16 | 94.87 |
| 6 | 84.11 | 79.44 | 83.18 | 79.44 | 82.24 | 82.71 | 94.02 | 96.01 | 94.87 | 94.59 | 95.16 | 94.59 |
| 7 | 83.18 | 82.71 | 86.91 | 81.31 | 82.71 | 78.51 | 95.72 | 96.58 | 93.16 | 95.16 | 94.87 | 94.87 |
| 8 | 82.24 | 84.11 | 82.24 | 82.24 | 84.11 | 82.24 | 96.58 | 97.15 | 95.16 | 94.02 | 94.59 | 93.16 |
| 9 | 82.24 | 81.77 | 85.51 | 78.51 | 82.71 | 79.44 | 94.87 | 95.72 | 94.59 | 95.16 | 94.59 | 94.02 |
| 10 | 82.71 | 78.51 | 85.04 | 79.44 | 78.51 | 82.24 | 94.59 | 96.58 | 93.45 | 94.87 | 94.59 | 93.16 |
| Min. | 79.9 | 77.57 | 81.31 | 78.51 | 78.51 | 78.51 | 94.02 | 95.72 | 92.88 | 92.87 | 93.16 | 93.16 |
| Max. | 85.04 | 84.11 | 86.91 | 82.24 | 84.11 | 82.71 | 96.58 | 97.15 | 95.16 | 95.16 | 95.16 | 94.87 |
| Avg. | 82.427 | 80.3 | 84.016 | 80.095 | 81.926 | 81.121 | 95.356 | 96.436 | 94.075 | 94.503 | 94.332 | 94.217 |

space dimension may be absolutely less than original feature space dimension. Table (5) shows the mapping spaces for LDA and PCA.

Table 5: Mapping spaces for LDA and PCA and their projected feature space dimension.

| Datasets | Dimension | Class | LDA | PCA |
|---|---|---|---|---|
| IRIS | 4 | 3 | $R^4 \rightarrow R^2$ | $R^4 \rightarrow R^2$ |
| WINE | 13 | 3 | $R^{13} \rightarrow R^2$ | $R^{13} \rightarrow R^6$ |
| GLASS | 9 | 6 | $R^9 \rightarrow R^5$ | $R^{94} \rightarrow R^5$ |
| IONOSPHERE | 34 | 2 | $R^{34} \rightarrow R^1$ | $R^{34} \rightarrow R^{10}$ |

In addition to LDA and PCA, obtained results have been compared with other reported classification rate that used same data sets in other literatures as shown in Table (6). Obtained results compared with other results, show the importance of work on data and features before using complex classifiers with high computational costs. In all data sets, both proposed methods (LIFC and NLIFC) provide high quality features, so a simple classifier such as minimum distance classifier can discriminates classes and presents easily acceptable classification rate: for Iris correct recognition arte is increased from % 94.66 to % 100and for Wine this rate is increased from % 76.96 to % 99.43. Also correct recognition rate reaches to % 86.91 for Glass and % 97.15 for Ionosphere.

Table 6: Maximum obtained result compared with maximum results of other classification systems which use same data sets..

| | Datasets | | | |
| --- | --- | --- | --- | --- |
| Methos | IRIS % | WINE % | Glass % | IONOSPHERE % |
| Minimum Distance | 94.66 | 76.96 | 73.36 | 86.61 |
| NLIFC[1] & Minimum Distance | 100 | 99.43 | 84.11 | 97.15 |
| LIFC[2] & Minimum Distance | 100 | 97.75 | 86.91 | 96.58 |
| LDA & Minimum Distance | 96.67 | 99.43 | 63.08 | 78.06 |
| PCA & Minimum Distance | 95.33 | 76.97 | 75.70 | 90.31 |
| NFL (nearest feature line) (Du & Chen (2007)) | 88.7 | 92.7 | 66.8 | 85.2 |
| 3-NN (3- nearest neighbor) (Du & Chen (2007)) | 94.70 | 95.50 | 72.00 | 84.60 |
| RNFLS (Rectified nearest feature line segment) (Du & Chen (2007)) | 95.30 | 97.20 | 72.00 | 94.30 |
| SFLS (Han et al. (2011)) | 96.00 | 96.10 | 70.10 | 92.40 |
| NNL (nearest neighbor line) (Han et al. (2011)) | 94.7 | 78.7 | 65.4 | 87.2 |
| Myopic algorithm (Ji & Carin (2007)) | n.a. | n.a. | n.a. | 92.28 |
| SVM (Seeger (2000)) | n.a. | n.a. | n.a. | 88.60 |
| Gaussian process (Kuss & Rasmussen (2006)) | n.a. | n.a. | n.a. | 92.01 |
| Nave Bayesian | 96.33 | 99.07 | 44.60 | n.a. |
| Decision tree (C4. 5) | 95.33 | 96.26 | 68.40 | 91.45 |
| Adamenn (Tahir et al. (2007)) | 97.00 | n.a. | 75.20 | 92.90 |
| TS/K-NN (Tahir et al. (2007)) | 96.70 | n.a. | 80.40 | 93.80 |
| LDics (Tahir et al. (2007)) | 94.67 | n.a. | 59.30 | 85.70 |
| LA-classifier (Zahiri (2008)) | 95.20 | 94.20 | 77.30 | n.a. |
| HHONC (Baesens et al. (2004)) | 99.77 | 99.43 | 64.08 | n.a. |
| Lat (Sanz et al. (2011)) | 97.33 | 91.52 | 61.20 | n.a. |
| IVFS_WI (Sanz et al. (2011)) | 96.00 | 93.79 | 59.34 | n.a. |
| IPS-classifier (Zahiri & Seyedin (2007)) | 91.33 | 95.30 | n.a. | n.a. |

[1] NLIFC = Non-Linear Intelligent Feature Combination

[2] Linear Intelligent Feature Combination

# 7 CONCLUSION

In order to design more efficient classification system extracting useful knowledge and features from data set is so important and helpful. In many cases, it is more reasonable to spend time and energy to analyze features instead of using more complex classifiers with high computational costs. In this paper intelligent feature combination is proposed to enhance the quality of features and then minimum distance classifier is used as a simple classifier to obtain results. Obtained results confirm that kind of combination method depends on nature of data set and its features. For some datasets using non-linear mapping system while reducing dimension of the feature space is useful and sometimes using linear mapping system while increasing the dimension of the feature space leads to design the classification system more efficiently. For Iris and Ionosphere using non-linear intelligent mapping system while reducing the dimension of feature space results correct recognition rates of %100 and % 97.15respectively. Using non-linear intelligent mapping while increasing dimension of feature space leads to obtain correct recognition rate of % 99.43 for Wine. It is so interesting that the best result for Glass obtains when features are combined by non-linear mapping without any change in dimension of feature space.

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
