# OpenReview forum: "ENRICHMENT OF FEATURES FOR CLASSIFICATION USING AN OPTIMIZED LINEAR/NON-LINEAR COMBINATION OF INPUT FEATURES"
_ICLR.cc/2018/Conference — Reject_

### Official Review · AnonReviewer1 · 2017-11-25
**Very poorly written paper that optimises a two layer neural network for feature generation using genetic algorithms.**

**Rating:** 1
**Confidence:** 5

**Review:**

The paper presents a method for feature projection which uses a two level neural network like structure to generate new features from the input features. The weights of the NN like structure are optimised using a genetic search algorithm which optimises the cross-validation error of a nearest neighbor classifier. The method is tested on four simple UCI datasets. There is nothing interesting or novel about the paper. It is not clear whether the GA optimisation takes place on the level of cross validation error estimation or within an internal validation set as it should have been the case. The very high accuracies reported seem to hint the latter, which is a serious methodological error. The poor language and presentation does not help in clearing that, as it does not help in general.

---

### Official Review · AnonReviewer3 · 2017-11-27
**Unclear what contributions are**

**Rating:** 3
**Confidence:** 4

**Review:**

This paper proposes using a feedforward neural network (FFNN) to extract intermediate features which are input to a 1NN classifier. The parameters of the FFNN are updated via a genetic algorithm with a fitness function defined as the error on the downstream classification, on a held-out set. The performance of this approach is measured on several UCI datasets and compared with baselines.
– The paper’s main contribution seems to be a neural network with a GA optimization for classification that can learn “intelligent combinations of features”, which can be easily classified by a simple 1NN classifier. But isn't this exactly what neural networks do – learn intelligent combinations of features optimized (in this case, via GA) for a downstream task? This has already been successfully applied in multiple domains eg. in computer vision (Krizhevsky et al, NIPS 2011), NLP (Bahdanau et al 2014), image retrieval (Krizhevsky et al. ESANN 2011) etc, and also studied comprehensively in autoencoding literature. There also exists prior work on optimizing neural nets via GA (Leung, Frank Hung-Fat et al., IEEE Transactions on Neural networks 2003). However, this paper claims both as novelties while not offering any improvement / comparison.
– The claim “there is no need to use more powerful and complex classifier anymore” is unsubstantiated, as the paper’s approach still entails using a complex classifier (a FFNN) to learn an optimal intermediate representation.
– The choice of activations is not motivated, and performance on variants is not reported. For instance, why is that particular sigmoid formulation used?
– The use for a genetic algorithm for optimization is not motivated, and no comparison is made to the performance and efficiency of other approaches (like standard backpropagation). So it is unclear why GA makes for a better choice of optimization, if at all.
– The primary baselines compared to are unsupervised methods (PCA and LDA), and so demonstrating improvements over those with a supervised representation does not seem significant or surprising. It would be useful to compare with a simple neural network baseline trained for K-way classification with standard backpropagation (though the UCI datasets may potentially be too small to achieve good performance).
– The paper is poorly written, containing several typos and incomplete, unintelligible sentences, incorrect captions (eg. Table 4) etc.

---

### Official Review · AnonReviewer2 · 2017-11-28
**This work does not have the quality to be accepted at ICLR.**

**Rating:** 2
**Confidence:** 3

**Review:**

The main issue is the scientific quality. What the authors call "intelligent mapping and combining system" for the proposed system is simply a fully connected neural network. Such systems have been largely investigated in the literature. The use of genetic algorithms has also been considered. Moreover, mapping features to some appropriate feature space has been widely investigated, including the choice of appropriate mapping. We didn't find anything "intelligent" in the proposed mapping.

There are many spelling and grammatical errors.

---

### Decision · Program_Chairs · 2018-01-29
**ICLR 2018 Conference Acceptance Decision**

**Decision:**

Reject

**Comment:**

The presented method essentially builds a model that remaps features into a new space that optimizes nearest-neighbor classification. The model is a neural network, and the optimization is carried out through a genetic algorithm.

Pros:
 - One major issue with neural network classification is that of a lack of explainability. Many networks are currently "black box" approaches. By moving to the optimization problem to that of building a feature space for nearest neighbor classification, one can, to a degree, alleviate the "black box" issue by providing the discovered nearest neighbor instances as "evidence" of the decision.
- Authors use established datasets.

Cons:
- Authors do not properly cite previous work, as brought up by reviewers. There is much literature on optimization of feature spaces (such as the entire field of metric learning), as well as prior approaches using genetic optimization. The originality and significance here is therefore not clear.